# Cold Storage Followed by Transplantation Induces Interferon-Gamma and STAT-1 in Kidney Grafts

**DOI:** 10.3390/ijms24065468

**Published:** 2023-03-13

**Authors:** Madison McGraw, David Miller, Sorena Lo, Nirmala Parajuli

**Affiliations:** 1Department of Pharmacology and Toxicology, University of Arkansas for Medical Sciences, Little Rock, AR 72205, USA; 2Division of Nephrology, University of Arkansas for Medical Sciences, Little Rock, AR 72205, USA

**Keywords:** kidney transplantation, interferon gamma, transcription factor, stat1, transcription factor, stat3, viaspan, cold ischemia

## Abstract

Cold storage (CS)-mediated inflammation, a reality of donor kidney processing and transplantation, can contribute to organ graft failure. However, the mechanisms by which this inflammation is perpetuated during and after CS remain unclear. Here, we examined the immunoregulatory roles of signal transducer and activator of transcription (STAT) family proteins, most notably STAT1 and STAT3, with our in vivo model of renal CS and transplant. Donor rat kidneys were exposed to 4 h or 18 h of CS, which was then followed by transplantation (CS + transplant). STAT total protein level and activity (phosphorylation) were evaluated via Western blot analysis and mRNA expression was tabulated using quantitative RT-PCR after organ harvest on day 1 or day 9 post-surgery. In vivo assays were further corroborated via similar analyses featuring in vitro models, specifically proximal tubular cells (human and rat) as well as macrophage cells (Raw 264.7). Strikingly, gene expression of IFN-γ (a pro-inflammatory cytokine inducer of STAT) and STAT1 were markedly increased after CS + transplant. STAT3 dephosphorylation was additionally observed after CS, a result suggestive of dysregulation of anti-inflammatory signaling as phosphorylated STAT3 acts as a transcription factor in the nucleus to increase the expression of anti-inflammatory signaling molecules. In vitro, IFN-γ gene expression as well as amplification of downstream STAT1 and inducible nitric oxide synthase (iNOS; a hallmark of ischemia reperfusion injury) was remarkably increased after CS + rewarming. Collectively, these results demonstrate that aberrant induction of STAT1 is sustained in vivo post-CS exposure and post-transplant. Thus, Jak/STAT signaling may be a viable therapeutic target during CS to mitigate poor graft outcomes when transplanting kidneys from deceased donors.

## 1. Introduction

As many as 70% of patients with end-stage renal disease (ESKD) await a kidney transplant while on dialysis [1], and the scarcity of transplantable kidneys contributes to ESKD mortality. Although kidneys from living donors are preferable due to more successful 5-year graft outcomes, kidneys obtained from deceased donors are the most common [2,3], constituting up to 70% of all transplants in the United States [1]. Globally, kidney transplants increased by 14.3% from 2020 to 2021, with only 38% of these consisting of living donors [4]. Cold storage (CS) solutions such as Viaspan are typically used to maintain the viability of deceased donor kidneys [5,6,7,8,9] until a recipient can be chosen. However, a well-known contributing factor to poor graft outcomes is prolonged exposure to CS, and we previously demonstrated in a rat model that CS exacerbates renal injury after transplantation [10,11,12,13]. After brain death, there is a robust inflammatory response that can damage deceased donor kidneys, which may also contribute to deceased donor graft vulnerability throughout CS [14,15]. Mitigating mechanisms of tissue injury during CS is of particular importance to decrease adverse graft outcomes and improve long-term graft survival, thus contributing to more successful therapeutic outcomes for patients with ESKD. The central purpose of this study was to identify CS-related immune targets and avenues of graft injury.

Inflammation during transplantation can be induced via various CS-related injuries to the donor kidney, and some of these mechanisms have yet to be fully elucidated [10,12,16]. For example, prolonged CS triggers the induction of MHC class II molecules and infiltration of T cells, monocytes, and macrophages within kidneys after transplantation in animal models [17,18,19,20,21]. We demonstrated in rat transplant models that CS combined with transplantation (CS + Tx) contributes to reduced proteasome function, increased mitochondrial dysfunction, activation of the complement system, increased inflammatory cytokines (such as TNF-α and IL-1β), and macrophage infiltration within kidneys [10,11,22,23,24,25,26]. However, the pathways by which these inflammatory molecules transmit signals and induce renal damage following transplantation are unknown.

Seven members of the STAT protein family are known in mammals (STAT1, STAT2, STAT3. STAT4, STAT5a, STAT5b, and STAT6), functioning as signal transducers between growth factor receptors on the plasma membrane and specific genes in the nucleus (membrane-to-nucleus signaling) [27,28]. Generally, STAT proteins reside in the cytoplasm in their latent form but are activated by the tyrosine kinase Jak [27,28,29,30,31]. Jak/STAT signaling begins with extracellular stimuli such as cytokines (e.g., IFN-γ) or growth factors interacting with their corresponding transmembrane receptors, leading to the trans-activation of receptor-bound Jaks [30,32]. Activated Jaks phosphorylate specific tyrosine residues of STATs, leading to STAT dimerization and translocation into the nucleus [33,34]. STATs then bind specific DNA sequences along with transcriptional cofactors to regulate the expression of genes involved in cell growth, differentiation, and survival, as well as apoptosis [29,34,35,36,37,38,39,40]. Nonspecific Jak signaling inhibition has been previously reported to suppress the expression of inflammatory cytokines and mitigate acute oxidative stress in renal tissue [40]. STAT1 and STAT3 additionally perform distinct functional roles within the context of inflammation [35,36,41,42,43,44,45,46,47]. Gene-knockout studies support opposing biological endpoints for STAT1 and STAT3. For example, STAT1 plays a predominant role in promoting inflammation, growth arrest, and apoptosis [35,36,37,38], while STAT3 is an essential gene that induces cell proliferation, promotes anti-inflammatory processes, and prevents apoptosis [42,48,49].

The induction of inflammatory cytokines and chemokines, known to stimulate many STAT protein cascades, is a hallmark of kidney graft failure [50,51]. While inflammatory cytokines such as TNF-α and IFNs are known to activate/induce STAT1 and STAT3 [39,52,53,54,55] we do not know if these cytokines transmit inflammatory signals through STAT1 and STAT3 during CS + Tx. However, our rat kidney transplant model demonstrated that an abnormal immune response is produced within kidneys after CS + Tx [10,11,22,23,24,25,26], suggesting that STAT1/3 signaling may be involved in promoting graft failure. Here, we report that CS + Tx increased IFN-γ expression within transplanted rat kidneys and activated STAT1 and STAT3 signaling cascades. The objective of this study was to pinpoint new therapeutic targets for the improvement of renal graft outcomes, and our results stand to provide new insights into the role of IFN-γ and Jak/STAT signaling during transplantation. However, further studies are needed to determine how this could best be implemented clinically.

## 2. Results

### 2.1. CS + Tx Activates STAT1 and STAT3

Previously, we showed that CS + Tx induces inflammation and extensive tissue injury within rat kidney grafts [11,22,23] and STAT transcription factors are known to transduce inflammatory signals from the extracellular environment to the nucleus [39,40,46,47,56]. Here, we tested the hypothesis that STAT1 and STAT3 are critical mediators of inflammation during CS + Tx. Renal extracts from rat transplant models were evaluated for phosphorylated (activated) and total levels of STAT1 and STAT3 factors (1-day post-surgery). SDS-PAGE and Western blot analyses showed an increase in total STAT1 protein in kidneys after 18 h CS + Tx compared to Sham (Figure 1A). As expected, we also observed an increase in phosphorylated (Y701) STAT1 in kidneys after 18 h CS + Tx (Figure 1A). We similarly analyzed the levels of total and phosphorylated (Y705 and S727) STAT3. CS + Tx increased STAT3 phosphorylation (Y705 and S727) in kidneys compared to control, but surprisingly, total STAT3 protein levels remained unchanged (Figure 1B).

### 2.2. CS Dysregulates STAT1 and STAT3 Proteins within Rat Kidneys

Because 18 h CS + Tx activated STAT1 and STAT3 in renal transplants, we further evaluated the effects of CS alone (4 h or 18 h) on STAT1 and STAT3 activation and expression in rat kidneys. SDS-PAGE and Western blot analyses identified a basal level of STAT1 protein (total) in untreated control kidneys; however, CS decreased the level of total STAT1 in a time-dependent manner (0 h CS > 4 h CS > 18 h CS) (Figure 2A). Y705 phosphorylation of STAT1 was not detected in untreated rat kidneys, and this remained unchanged after CS (4 h or 18 h) treatment (Figure 2A). Levels of total STAT3 were unchanged after CS (4 h and 18 h) (Figure 2B), and basal levels of STAT3 phosphorylation (S727 and Y705) were observed in untreated kidneys (Figure 2B). Interestingly, S727 phosphorylation of STAT3 decreased in kidneys after 4 h CS, and this effect was further enhanced at 18 h CS, whereas Y705 phosphorylation of STAT3 decreased only after 18 h CS, (Figure 2B). These results suggest that CS alone affects STAT total and phosphorylated levels.

### 2.3. CS + Tx Induces Expression of STAT1, STAT3, and Inflammatory Cytokine Genes

Because CS + Tx (18 h CS, 1-day post-surgery) increased the total STAT1 protein level compared to STAT3, which remained unchanged, we evaluated the effect of CS + Tx on the expression of STAT1 and STAT3 genes in kidneys. Sham and 18 h CS kidneys were used as controls. Quantitative real-time PCR showed a significant increase in STAT1 and STAT3 mRNA in rat kidney tissues after 18 h CS + Tx, and in the case of STAT1, this increase was >3-fold (Figure 3).

Inflammatory cytokines transduce signals primarily by activating Jak/STAT pathways [39,40,46,47,56], and IFN-γ, a major inflammatory cytokine, is a potent activator of STAT1 [45]. STAT1 activation could induce inducible nitric-oxide synthase (iNOS), a hallmark of renal ischemia–reperfusion injury and oxidative stress. Similarly, IL-12α potentiates IFN-γ-mediated signal transduction by inducing the secretion of IFN-γ in different cell types [57]. Therefore, we evaluated the effects of 18 h CS + Tx on the expression of IFN-γ, IL-12α, iNOS, and endothelial nitric oxide synthase (eNOS) genes in renal grafts. Quantitative real-time PCR revealed significantly higher expression of IFN-γ, IL-12α, and iNOS (fold change vs. sham: 31-, 38-, and 16-fold, respectively), but not eNOS, in kidneys after CS + Tx compared to sham and CS-only controls (Figure 4).

### 2.4. CS Plus Rewarming Dysregulates the Expression of STAT1, STAT3, and iNOS Genes in Rat Proximal Tubular Cells

Previously, we reported that 18 h of CS followed by 6 h of rewarming (CS + RW) led to ~24% NRK (rat proximal tubular) cell death [11]. Here, we attempted to verify the expression and opposing biological effects of STAT1 and STAT3 genes in the context of rewarming using our in vitro renal CS models. Expression of both STAT1 and STAT3 in NRK cells decreased after 18 h CS (Figure 5A). CS + RW significantly decreased STAT1 expression in NRK cells (Figure 5A). Interestingly, STAT3 expression returned in NRK cells after CS + RW (Figure 5A), suggesting the decrease was caused by CS and not by CS + RW. Because iNOS is a downstream target of STAT1 signaling [55,58], we assayed iNOS expression in NRK cells after CS or CS + RW. Quantitative real-time PCR showed that CS did not affect iNOS expression, but CS + RW significantly increased iNOS expression (Figure 5B), suggesting that iNOS induction is associated with STAT1 signaling.

### 2.5. CS Decreases STAT3 Phosphorylation in Rat Proximal Tubular Cells

Because CS and CS + RW dysregulated STAT1 and STAT3 expression, we sought to determine the effects of 18 h CS ± RW on the levels of total and phosphorylated STAT1 and STAT3 in NRK and HK-2 (human proximal tubular) cells. Neither CS nor CS + RW affected the levels of total or phosphorylated (Y705) STAT1 (Figure 6A). Consistent with our in vivo results, 18 h CS decreased the levels of phosphorylated STAT3 (Y705 and S727) in NRK cells, and CS + RW sustained this decrease in STAT3 phosphorylation (Figure 6B). Neither CS nor CS + RW affected the level of total STAT3 protein (Figure 6B), consistent with our in vivo findings.

As in NRK cells, HK-2 cells CS ± RW did not affect the levels of total or phosphorylated STAT1 (Figure 7A). Consistent with our data for NRK cells and the in vivo model, 18 h CS decreased STAT3 phosphorylation (Y705) in HK-2 cells. Phosphorylation of STAT3 increased slightly after CS + RW in HK-2 cells, but not to the level of the control; also as shown in vivo (Figure 7B), the total levels of STAT3 did not change.

### 2.6. IFN-γ, but Not TNF-α, Activates STAT1 and STAT3 in Raw 264.7 Cells

Inflammatory cytokines such as IFN-γ and TNF-α are potent activators of STAT1 and STAT3 in various cells, including macrophages. Because we observed induction of these inflammatory cytokines (Figure 4) as well as macrophage infiltration [22] within rat kidneys after CS + Tx, we sought to determine the effects of IFN-γ or TNF-α on STAT transcription factor activation within macrophages in the context of CS ± RW. Raw 264.7 macrophages were treated with CS ± RW, IFN-γ, or TNF-α, and the cell lysates were assayed with Western blot. As anticipated, IFN-γ increased both phosphorylated and total STAT1 and STAT3 in Raw 264.7 macrophages. (Figure 8A,B). Surprisingly, TNF-α did not affect the levels of phosphorylated or total STAT1 or STAT3 (Figure 8A,B). These results suggest that IFN-γ is the inducer of STATs and this induction requires some other factors within the kidney microenvironment.

### 2.7. Activation and Induction of STAT1 and STAT3 Persist in Renal Allografts at Day 9 after CS + Tx

We sought to determine if CS time and transplant duration is an important determinant of phosphorylation and induction of STAT1 and STAT3. Renal extracts from rats that underwent 4 h or 18 h CS + Tx were assayed for total and phosphorylated STAT1 and STAT3 at day 9 post-surgery; the sham group was used as a control. Western blot indicated that both 4 h CS + Tx and 18 h CS-Tx increased total and phosphorylated STAT1 (Y701) at day 9 post-surgery (Figure 9A). Likewise, total and phosphorylated (S727) STAT3 levels remained elevated at day 9 (Figure 9B).

## 3. Discussion

In this study, we demonstrated that CS exposure dephosphorylates STAT3 within donor rat kidneys. Subsequently, we showed that CS + Tx activates STAT1 and STAT3 factors and increases STAT1 protein levels within transplanted kidneys. Phosphorylation of STAT1 at Y701 is required for transcriptional activation of STAT1 [45,56,57,58,59,60,61], and this activation induces cell cycle/growth arrest and promotes apoptosis [62,63,64,65,66]. Similarly, phosphorylation of STAT3 at Y705 is required for transcriptional activation of STAT3 [61,67,68], and in most instances, this activation induces cell growth/proliferation that promotes cell survival [69,70,71]. Upon activation, STAT1 or STAT3 localize in the nucleus and promote transcription of their downstream target genes, including STAT1 or STAT3, respectively [39,40]. We did not observe STAT1 phosphorylation (Y701) within kidneys under untreated, CS (4 h or 18 h), or ATx conditions, but phosphorylation increased after CS + Tx. Interestingly, the transcriptional activity of STAT3 (Y705 phosphorylation) decreased after 18 h CS, but reappeared after CS + Tx when compared to sham/control kidneys. Y705 phosphorylation also reappeared after CS + RW in vitro (although this phosphorylation was diminished when compared to untreated cells), suggesting that STAT3 transcriptional activity was only partially restored in proximal tubules after CS + RW. Unlike STAT1, our CS + Tx and CS + RW models did not elevate STAT3 protein levels in kidneys or renal proximal tubule cells, respectively, suggesting there is a sustained increase in STAT1 activation. Because CS + Tx exacerbates renal graft injury when compared to ATx [10,11], we hypothesize that this sustained STAT1 activation and induction might have promoted CS + Tx-mediated injury.

While we observed an increase in STAT1 phosphorylation after CS + Tx in vivo (Figure 1), we did not observe a similar increase in STAT1 phosphorylation after CS + RW in vitro. Similarly, there were discrepancies in STAT1 levels between in vivo (Figure 2) and in vitro (Figure 5) systems. The in vivo kidney microenvironment contains an intricate web of inciting factors (including cytokines and chemokines) that influence a multitude of cellular processes, and this diversity cannot be fully replicated in vitro. Local paracrine signaling is a physiologic process integral to the kidney, fostering a collective and protective renal tissue response to regulate overall kidney function in response to various systemic stimuli [72]. Additionally, the continued upregulation of STAT1 activation in kidney grafts after CS + Tx at the 9-day post-surgery timepoint lends further support to our earlier hypothesis. A major limitation is that we could not dissect the role of STATs or cytokines (e.g., IFN-γ) due to the unavailability of transgenic/knockout rats. While the inhibitors are available, the use of pharmacological inhibitors may not produce desirable and reproducible effects as this would further amplify the stress that is produced by CS + Tx in rats. Another limitation of this study was that due to the complexity of rodent transplant surgery, we performed these experiments only with male rats. In the future, we will include both sexes in our studies.

STAT1 and STAT3 have been shown to contribute to kidney damage during warm ischemia–reperfusion injury (IRI) in mice [31,73]. Genetic knockout of STAT1 in mice attenuated acute tubular necrosis and decreased macrophage infiltration after IRI [73]. Similarly, IRI increased STAT3 in kidneys, and pharmacologic inhibition of STAT3 with Stattic attenuated acute kidney injury and decreased macrophage infiltration in mice [31]. Conversely, in other organs such as the heart, it has been shown that STAT3 could play a protective role during myocardial IRI [74]. One proposed mechanism of protection is STAT3 activation and localization in mitochondria during myocardial IRI—in this case, STAT3 inhibits the opening of mitochondrial permeability transition pores, and this confers resiliency against IRI [74,75]. Several studies reported the localization of STAT3 to mitochondria, where it functions to regulate oxidative phosphorylation and reactive oxygen species [76,77,78,79]. STAT3 phosphorylation at Y705 and S727 is required for cell proliferation and mitochondrial gene transcription, respectively [80]. This has been explored by other investigators, as Yang J. et al. and Wakahara R. et al. reported that S727 phosphorylation leads to reduced transcriptional activity of STAT3 via specific dephosphorylation of Y705 [81]. S727 phosphorylation also stabilizes the STAT3–DNA-binding complex in vitro [82]. Strikingly, in our rat kidney CS models, STAT3 S727 phosphorylation decreased after only 4 h of CS, and this decrease persisted for 18 h of CS; STAT3 S727 phosphorylation reappeared after CS + Tx. We previously reported decreased mitochondrial membrane potential, reduced mitochondrial respiration complex activity, and impaired mitochondrial fission/fusion after CS + Tx [10,11]. Taken together, these results suggest that decreased STAT3 phosphorylation (Y705 and S727) during CS could decrease mitochondrial respiration in donor kidneys. Similarly, our results indicating partial activation of STAT3 after CS + Tx or CS + RW imply that STAT3 activation is deficient and potentially insufficient to protect against CS + Tx-mediated mitochondrial respiratory complex dysfunction, thus leading to graft injury. Future studies are warranted to evaluate the role of STAT3 (mitochondrial- and nuclear-localized) in mitochondrial and renal injury during CS ± Tx.

The dephosphorylation of STAT3 (Y727 and Y705) within rat kidneys during CS was a unique observation. CS-mediated dephosphorylation of STAT3 in donor kidneys suggests a potential role for STAT3-associated phosphatases, including protein phosphatase 2A (PP2A), which is upregulated during renal IRI. PP2A dephosphorylates Thr172 of AMP-activated protein kinase α (AMPKα) [83], a kinase known to phosphorylate mitochondrial fission factor during energetic stress [84,85]. PP2A can also dephosphorylate S727 of STAT3 [82,86], and it inhibits the formation of serine-dependent STAT1/IFN-γ activation factor-DNA binding complexes in vitro [87]. At this juncture, the specific mechanisms of CS-related STAT3 dephosphorylation are unclear. Further studies are needed to characterize potential CS-mediated PP2A activation and its role in STAT3 dephosphorylation, as well as subsequent mitochondrial dysfunction, within donor rat kidneys.

In addition to mechanisms of STAT phosphorylation, it is worthwhile to consider STAT1/3 transcriptional targets as key mediators of renal injury. Of note, IRI activates nitric oxide synthase (NOS) isoforms, and inducible NOS (iNOS) is thought to produce detrimental effects during IRI, whereas endothelial NOS (eNOS) is thought to protect against IRI [88,89,90,91,92,93]. iNOS is a STAT1-target gene [55,58], and STAT3 suppresses iNOS expression via interaction with NFkB [94,95]. Previously, we showed an increase in peroxynitrite levels (an oxidant) after CS + Tx and CS + RW [96,97]. Here, we similarly observed induction of iNOS after CS + Tx and CS + RW, suggesting that IRI promotes nitrosative stress within kidneys and, specifically, proximal tubular cells. Indeed, inhibiting iNOS activity reduced NO-mediated renal dysfunction following renal IRI [98]. These results suggest that CS + Tx-mediated STAT1 activation could upregulate iNOS to contribute to oxidative stress and renal injury.

We previously showed that CS + Tx induced inflammatory genes (Ccl2/MCP-1, TNF-α, IL-1β, and CD11b/c), increased complement activation, and promoted CD68+ macrophage infiltration [22,23] within renal grafts as soon as 1-day post-surgery. The levels of 2 inflammatory cytokines, IFN-γ and IL-12α, are increased in response to tissue damage during renal IRI [99]. Interferon-gamma (IFN-γ) is a potent activator of STAT1 [45], and in the absence of STAT1, IFN-γ exacerbates the activation/induction of STAT3 [100]. IL-12α potentiates IFN-γ-mediated signal via induction of IFN-γ secretion within different cell types [57]. As anticipated, CS + Tx induced IFN-γ and IL-12α expression in renal grafts, both of which regulate IFN-γ expression [101]. IFN-γ mediates its pro-inflammatory signal by activating STAT1 transcription factors and through its additional macrophage-stimulating properties [102,103]. Moreover, IFN-γ treatment induces Ccl2/MCP-1 in mesangial cells [104,105,106]. IFN-γ-mediated induction of MCP1 could be STAT1-dependent or -independent [107,108], and there is a noteworthy functional relationship between MCP-1 and STAT3 activation. Specifically, following treatment of vascular smooth muscle cells with 15-lipoxygenase, src-STAT3 signaling mediated expression of MCP-1 [109]. In other studies, the activation of MCP-1 activated the JAK2/STAT3 pathway in the Mono Mac1 monocytic cell line [110], suggesting that both MCP-1 and STAT3 are engaged in a positive feedback loop. In our renal models, CS + RW or CS + Tx increased both MCP-1 and STAT3 expression and also increased STAT3 phosphorylation (Y705 and S727). Future work should elucidate the functional relationship of MCP-1 and STAT3 activation in the context of renal CS + Tx.

Immune cells are the primary source of inflammatory cytokines that promote STAT1 and STAT3 activation in various cell types [70,111]. Here, we used an in vitro system to gain further insight into STAT1 and STAT3 activation in renal proximal tubular cells and macrophages, complementing our in vivo transplant models. In addition to the cytokines IFN-γ and IL-12α (Figure 4), CS + Tx also induces IL-1β, TNF-α, and Ccl2 in rat kidneys [22,23]. Similar studies were observed using human kidney transplants [112]. We observed that stimulation with IFN-γ has the potential to activate and induce expression of both STAT1 and STAT3 factors in Raw 264.7 cell lines. As expected, IFN-γ also potentiated the activation and induction of STAT1 in NRK cells. TNF-α, a pro-inflammatory cytokine that did not induce STAT1/3 in our experiments with Raw 264.7, has been shown to phosphorylate S727 of STAT1—a process required for complement gene transcription via Y701 phosphorylation of STAT1 during IFN-γ signaling [113]. Future studies should target the Y701 or S727 phosphorylation of STAT1 during CS and CS + RW/Tx to further examine if TNF-α synergizes IFN-γ-mediated STAT1 transcriptional activation (Y705 STAT1).

While STAT1 plays an important role in propagating signals from inflammatory cytokines/receptors (e.g., IFN-γ), STAT3 engages predominantly in transmitting anti-inflammatory signals and signals involved in the maturation/regulation of the innate and adaptive immune systems [27,28,29,30,31,34,35,68,111,114]. Importantly, STAT3 mediates the anti-inflammatory effects of IL-10 [115] and G-CSF signaling during emergency granulopoiesis [31,114,116]. STAT3 also plays a key role in suppressing signal transduction mediated by Toll-like receptors (TLRs) in mature phagocytic cells [117,118]. Accordingly, mice with STAT3 deficiency in macrophages and neutrophils displayed increased levels of circulating pro-inflammatory cytokines and diminished IL-10 levels [118,119]. In addition, small-molecule-mediated inhibition of STAT3 signaling downregulated Ccl2/Ccl12 chemokines, which impaired neurofibroma macrophage populations [120]. Thus, STAT3 has a pivotal role in maintaining a signaling balance during the innate immune response. Given our findings that CS decreased STAT3 phosphorylation and that CS + Tx dysregulated STAT3 transcriptional activity, we hypothesize that STAT1 activity (rather than STAT3) is predominantly involved in inflammatory signaling and macrophage infiltration within rat kidney grafts after transplantation.

In summary, we showed that CS impaired the phosphorylation of STAT3 at Y705 and S727 in donor kidneys, and STAT3 transcriptional activity was only partially restored in grafts after CS + Tx. IFN-γ and iNOS gene expression and STAT1 activity (Y701 phosphorylation) and protein levels increased after CS + Tx, suggesting that IFN-γ signaling activates the STAT1 pathway to induce iNOS within kidneys during CS + Tx. In vitro experiments using NRK cells and Raw 264.7 macrophages confirmed that IFN-γ contributed to STAT1 activation in renal tubular cells as well as STAT1/3 activation in macrophages. Altogether, these results suggest that IFN-γ-mediated activation of STAT1 within kidneys could induce renal damage following CS + Tx.

## 4. Materials and Methods

### 4.1. Animals

Male Lewis or Fischer rats (200–250 g) were obtained to function as transplant donors, and transplant recipients were male Lewis rats. All animal protocols were approved by the Institutional Animal Care and Use Committee at the University of Arkansas for Medical Sciences, and all animal experiments were performed in compliance with institutional and NIH guidelines.

### 4.2. Rat Surgery

Transplant surgeries were performed as described in previous publications [10,11,22,23,26].

#### 4.2.1. Cold Storage Plus Transplant (CS + Tx) Surgery

Donor surgeries were performed via CS solution flushing and removal of left and right kidneys from Lewis (Charles River Laboratories, Wayne, PA, USA) (1-day post-surgery) or Fischer (Charles River Laboratories, USA) (9-day post-surgery) rats. Excised kidneys were stored in CS solution (Viaspan) for 4 h or 18 h at 4 °C. Right kidneys constituted CS exposure-only groups; left kidneys were reserved for transplantation. Recipient surgeries were performed on anesthetized Lewis rats; the native left kidney was excised, and donor kidneys were attached via anastomosis of the renal blood supply, keeping the ischemic period under 45 min. The ureter was additionally anastomosed using a 5 mm PE-50 polyethylene stent. Native right kidneys were removed from recipients either during initial transplantation (for 1-day post-surgery time points with donor Lewis rats) or on day 7 after the initial transplantation (for 9-day post-surgery time points with donor Fischer rats). Post-operative care was identical for all animals, with re-hydration via a subcutaneous (SC) injection of 0.9% *w*/*v* NaCl and placement in an appropriate heated environment (heating pad). Pain management consisted of buprenorphine (2 mg/kg, SC) at appropriate intervals; groups reserved for the 9-day post-surgery survival time point also received cyclosporine A (1.5 mg/kg/day, SC) for immunosuppression. Surgery survival rates were >95%. Transplanted kidneys were collected at 1 or 9 days post-surgery and were assigned to CS + Tx groups (*n* = 3). Collected kidneys were processed and stored at −80 °C for further analysis.

#### 4.2.2. Autotransplant (ATx) Surgery

The purpose of autotransplant groups (ATx) was to compare the effect of CS exposure vs. the effect of transplantation alone, and ATx procedures were performed similarly to CS + Tx groups. For ATx (*n* = 3), left nephrectomies were performed and organs were flushed with saline before re-transplantation back into native animals; all right kidneys were removed during the initial ATx procedure. ATx kidneys were harvested after 1-day of reperfusion post-surgery.

#### 4.2.3. Sham Surgery

Rats underwent the same procedure for right nephrectomy without renal transplantation (sham operation). The right kidney was saved as an untreated control kidney (*n* = 3). The left kidney was harvested 1 or 9 days later and served as the sham group (*n* = 3).

### 4.3. Cell Culture and Treatment

Normal rat kidney proximal tubular cells (NRK-52E; ATCC No. CRL-1571), human kidney proximal tubular cells (HK-2; ATCC No. CRL-2190), and monocyte/macrophage-like cells (Raw 264.7; ATCC No. TIB-71) were maintained at (37 °C) in the appropriate growth media (NRK: DMEM plus 5% fetal calf serum and 1% penicillin/streptavidin, 5% CO2; HK-2: DMEM plus 5% fetal calf serum and 1% penicillin/streptavidin, 5% CO_2_; and Raw: DMEM plus 10% fetal calf serum and 1% penicillin/streptavidin, 5% CO_2_).

#### 4.3.1. CS Plus Rewarming (CS + RW) Treatment

NRK and HK-2 cells were exposed to CS at 70% confluence as reported previously [9]. CS condition cells were washed briefly in cold PBS (4 °C) before being incubated with cold UW solution (4 °C) for 18 h. Rewarming was initiated by washing the cells with cold PBS and then adding cold growth medium (4 °C) before incubating cells in growth medium at 37 °C for 6 h.

#### 4.3.2. Cytokine Treatment

Raw 264.7 cells were cultured to 60% confluence and were incubated in growth medium containing IFN-γ (10 ng/mL; R&D Systems) or TNF-α (10 ng/mL; R&D) at 37 °C for 24 h.

### 4.4. SDS-PAGE and Western Blot Analysis

Renal lysates were prepared and analyzed via SDS-PAGE Western blot as described previously [23]. Membranes were incubated with antibodies to STAT1 (1:1000; Cell Signaling Technology [CST], Danvers, MA, USA, #14994), phospho-STAT1 (Y701) (1:1000; CST #7649), STAT3 (1:1000; CST #30855), phospho-STAT3 (Y705) (1:1000; CST #9145), or phospho-STAT3 (S727) (1:1000; CST #9134), as well as β-actin (loading control; 1:1000; Sigma-Aldrich, St. Louis, MO, USA, #A5441). Probed membranes were washed 3 times with tris-buffered saline containing 0.01% Tween-20 (TBS-T) and incubated with horseradish peroxidase-conjugated secondary antibodies (1:30,000; Seracare KPL) before membranes were assayed for enhanced chemiluminescence (Thermo Fisher Scientific, USA). Densitometry was performed using AlphaEase FC 8900 software (Alpha Innotech, San Leandro, CA, USA). For all SDS-PAGE Western blots of kidney or renal cell extracts, the densitometry ratio of pSTAT1 to total STAT1 or pSTAT3 to total STAT3 was considered for statistical analysis. For STAT1 or STAT3 total protein, the densitometry ratio of total STAT1/3 to β-actin was considered for statistical evaluation.

### 4.5. Quantitative Real-Time PCR

Cell/tissue isolated RNA was obtained using the RNeasy kit (Qiagen) and reverse-transcribed with Superscript III (Invitrogen). Quantitative real-time PCR was performed for 45 cycles with the SYBR green PCR Kit [Quiagen] (Beverly, MA, USA) [23]. Reaction conditions were as follows: 95 °C for 30 s, 58 °C for 30 s, and 72 °C for 1 min. Amplification of the target was normalized to the amplification of TATA box binding protein (TBP) and to the levels of an appropriate control using the delta Ct method [55]. The specific primer sequences (RealTimePrimers.com, accessed 2018-2022) used were as follows: STAT1: Forward, 5′-TCT TGG GAC GTA GCT GAG TG-3′ and Reverse, 5′-GCC GGT AAG AGC TGA GAT TC-3′; STAT3: Forward, 5′-GAC CGT CTG GAA AAC TGG AT-3′ and Reverse, 5′-ACA GAT CCA CGA TCC TCT CC-3′; iNOS: Forward, 5′-TCA GGC TTG GGT CTT GTT AG-3′ and Reverse, 5′-GGAAACCATTTTGATGCTTG-3′; eNOS: Forward, 5′-AGC ACT TGG AAA ATG AGC AG-3′ and Reverse, 5′-CAC TGC ATT GGC TAC TTC CT-3′; IFN-γ: Forward, 5′-CCA AGT TCG AGG TGA ACA AC-3′ and Reverse, 5′-ACT CCT TTT CCG CTT CCT TA-3′; IL-12α: Forward, 5′-CTG CCT CCA CAA AAG ACT TC-3′ and Reverse, 5′-GAT TCA GAG ACC GCA TTA GC-3′; and TBP: Forward, 5′-CGA TAA CCC AGA AAG TCG AA-3′ and Reverse, 5′-AGA TGG GAA TTC CAG GAG TC-3′.

### 4.6. Statistical Analysis

Mean ± standard error of the mean (SEM) for all results is shown as appropriate (GraphPad Prism software, version 9). A 1-way ANOVA—followed by Brown–Forsythe and Welch’s correction for multiple group comparisons—was used for data analysis of more than 2 groups; an unpaired Student’s *t*-test was used when comparing differences between 2 groups (such as Control vs. CS) at a 95% level of confidence. Differences with *p* < 0.05 were taken as statistically significant. Untreated kidneys and those treated with CS alone were both harvested from healthy rats. Sham kidneys served as a negative control for all transplant group kidneys, as each rat within these treatment groups underwent a right nephrectomy.

## 5. Conclusions

During the process of kidney transplantation, donor organs will be exposed to cold-storage (CS) solution, an event known to exacerbate ischemia–reperfusion injury and contribute to graft failure. Therefore, CS-mediated changes in protein induction have a potential role in priming cells for injury and are recognized as important targets for therapeutic intervention. Changes in the expression of certain transcription factors, such as STAT, are of note because as they may be responsible for driving cellular growth and inflammatory states. The present study sheds light on the activity of two prominent STAT factors, STAT1 and STAT3, in response to CS, using syngeneic and allogeneic rat kidney transplant models. For the first time, we demonstrated an aberrant and sustained induction of STAT1 after CS + transplant for up to 9 days post-surgery. IFN-γ, a known and potent inducer of STAT1, was similarly elevated after exposure to CS. We also demonstrated a CS-mediated increase in STAT3 dephosphorylation in vivo, the true implications of which require further study. Collectively, these results highlight a key shift in cellular signaling after CS + transplant that may have a role in reperfusion-related injury to organ grafts after transplant. Future studies should explore the mechanisms by which STAT signaling alters gene expression profiles in cells after exposure to CS and delineate pathways that make organs susceptible to reperfusion injury. Our findings suggest that therapeutics targeting the Jak/STAT signaling cascade during CS and the critical balance between STAT1 and STAT3 induction could improve clinical outcomes in patients requiring kidney transplants.

## Figures and Tables

**Figure 1 ijms-24-05468-f001:**
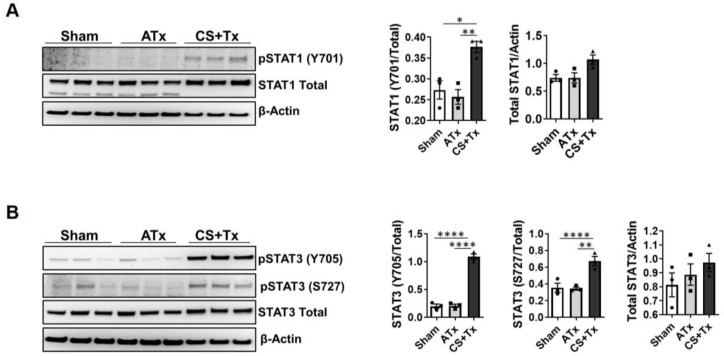
Activation/induction of STAT after cold storage and transplantation. Experimental groups post-1 day of reperfusion; healthy Sham, auto-transplant (ATx) without CS, and transplant after 18 h CS exposure (CS + Tx) shown. (**A**,**B**) Homogenized kidneys lysed with RIPA (30 μg) were loaded for SDS-PAGE Western assays and figures depict densitometric analysis of (**A**) Y701-phosphorylated/total STAT1, as well as (**B**) Y705- and S727-phosphorylated/total STAT3. Representative blots from 3 independent experiments show mean ± SEM (bars, *n* = 3). Data were computed using 1-way ANOVA followed by Brown–Forsythe and Welch’s corrections for multiple group comparisons; *p* < 0.05 is considered significant. *, *p* < 0.05; **, *p* < 0.01, ****, *p* < 0.001. Each solid circle, square, or triangle represent individual animals within the group.

**Figure 2 ijms-24-05468-f002:**
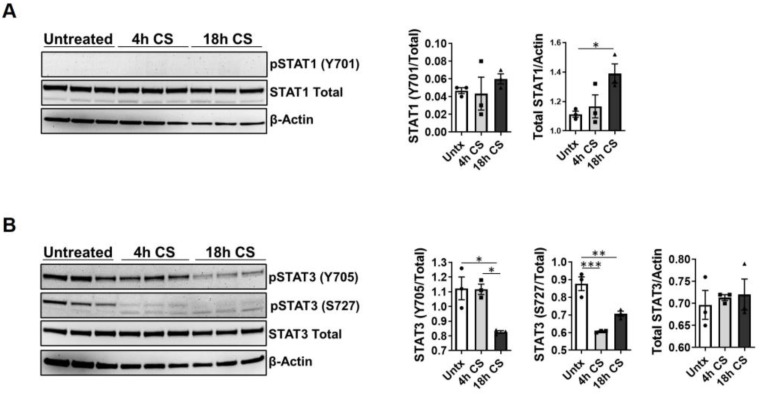
Cold storage decreases phosphorylated STAT3 in donor kidneys. Three experimental groups were considered: Untreated control, 4 h CS, and 18 h CS; in CS groups, kidneys were isolated, flushed, and stored in University of Wisconsin CS solution for the indicated time; (**A**,**B**) Renal extracts (30 μg) were prepared using RIPA and assayed via SDS-PAGE Western blot (representative blots are shown) as previously: (**A**) Y701-phosphorylated/total STAT1, (**B**) Y705- and S727-phosphorylated/total STAT3. Data are the mean ± SEM (bars, *n* = 3). Differences between group means were compared with 1-way ANOVA followed by Brown–Forsythe and Welch’s corrections for multiple group comparisons; *p* < 0.05 was considered significant. *, *p* < 0.05; **, *p* < 0.01, ***, *p* < 0.001. Each solid circle, square, or triangle represent individual animals within the group.

**Figure 3 ijms-24-05468-f003:**
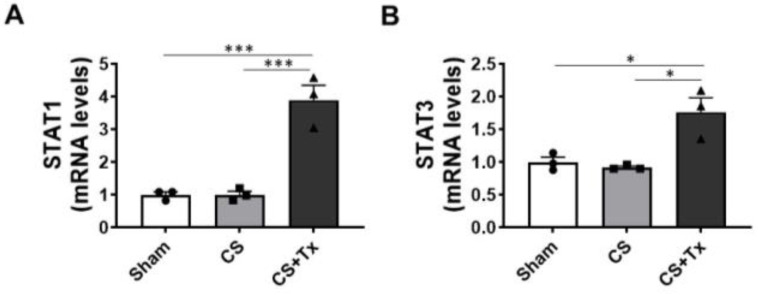
Cold storage plus transplantation induces STAT1 and STAT3 genes. Sham, 18 h CS, and 18 h CS + Tx groups underwent mRNA extraction and quantitative real-time PCR (SYBR Green) to assay the expression of (**A**) STAT1 and (**B**) STAT3. Data are the mean ± SEM (bars, *n* = 3), compared via 1-way ANOVA followed by Brown–Forsythe and Welch’s corrections for multiple group comparisons; *p* < 0.05 considered significant comparing Sham vs. CS + Tx and CS vs. CS + Tx groups. *, *p* < 0.05; ***, *p* < 0.001. Each solid circle, square, or triangle represent individual animals within the group.

**Figure 4 ijms-24-05468-f004:**
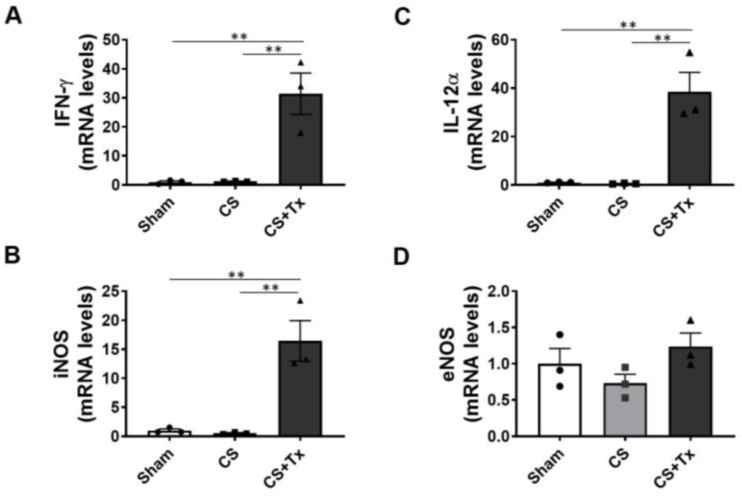
Cold storage plus transplantation induces inflammatory cytokine and nitric-oxide synthase genes. Quantitative real-time PCR (SYBR Green) was performed with rat renal mRNA extracts (as in previous figures) to assay expression of (**A**) IFN-γ, (**B**), IL-12α, (**C**) inducible nitric-oxide synthase (iNOS), and (**D**) epithelial nitric-oxide synthase (eNOS). Data are the mean ± SEM (bars, *n* = 3). 1-way ANOVA followed by Brown–Forsythe and Welch’s corrections for multiple group comparisons were used for analysis; *p* < 0.05 was considered significant for Sham vs. CS + Tx and CS vs. CS + Tx groups. **, *p* < 0.01. Each solid circle, square, or triangle represent individual animals within the group.

**Figure 5 ijms-24-05468-f005:**
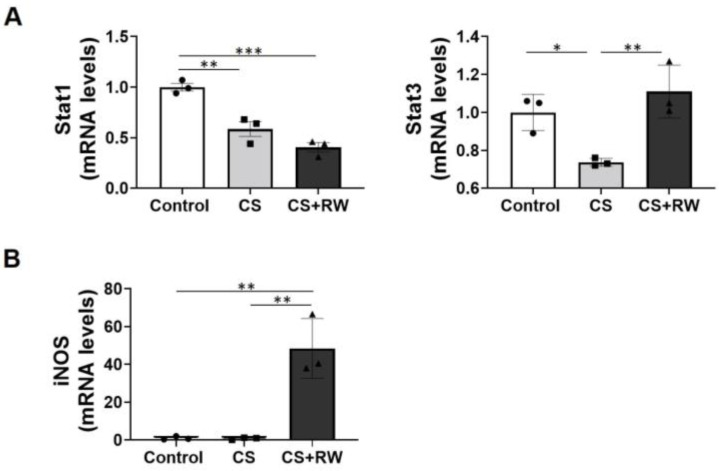
In vitro cold storage and/or rewarming dysregulates the expression of STAT1, STAT3, and iNOS in rat renal cells. NRK cells were exposed to either 18 h CS or CS + RW (re-warming; 6 h). Quantitative real-time PCR (SYBR Green) was performed to assess (**A**) STAT1, STAT3, and (**B**) iNOS after CS or CS + RW. Data are the mean ± SEM (bars, *n* = 3) and comparisons were made via 1-way ANOVA followed by Brown–Forsythe and Welch’s correction for multiple group comparisons; *p* < 0.05 was considered significant between Control vs. CS + RW and CS vs. CS + RW groups. *, *p* < 0.05; **, *p* < 0.01, ***, *p* < 0.001. Each solid circle, square, or triangle represent individual animal within the group.

**Figure 6 ijms-24-05468-f006:**
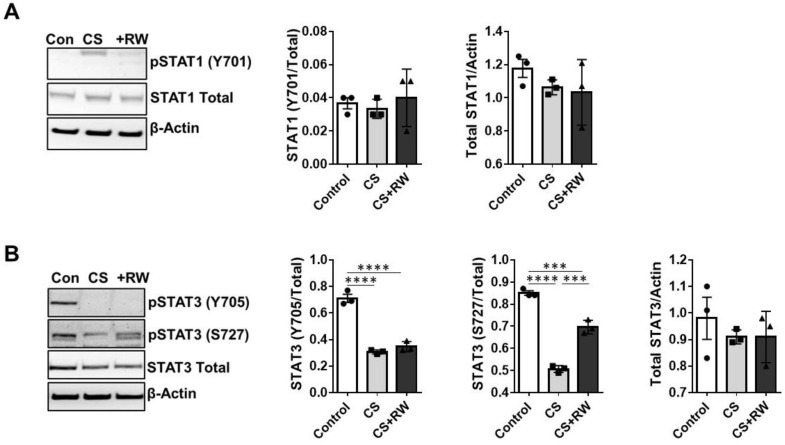
In vitro cold storage plus rewarming reduces STAT3 phosphorylation in rat renal cells. (**A**,**B**) NRK cells were exposed to 18 h CS or 18 h CS plus 6 h rewarming (CS or CS + RW), and renal extracts were prepared using RIPA buffer. Renal cells extract protein (20 μg) was resolved with SDS-PAGE and analyzed by Western blot for phosphorylated and total levels of (**A**) STAT1 and (**B**) STAT3 after CS or CS + RW. Data shown are the mean ± SEM (bars, *n* = 3) with representative blots. Group means were compared with 1-way ANOVA followed by Brown–Forsythe and Welch’s corrections for multiple group comparisons; *p* < 0.05 was considered significant. ***, *p* < 0.01, ****, *p* < 0.0001. Each solid circle, square, or triangle represent each experiment analysis (biological replicates) within the group.

**Figure 7 ijms-24-05468-f007:**
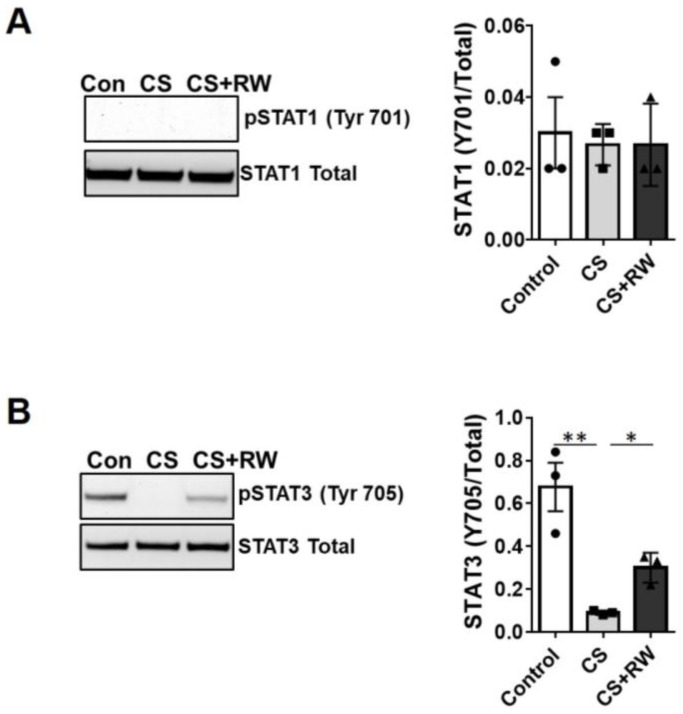
In vitro cold storage plus rewarming reduces STAT3 phosphorylation in human renal cells. (**A**,**B**) HK-2 cells were cultured and underwent 18 h CS or 18 h CS plus 6 h rewarming (CS + RW or +RW). Extracts were prepared using a RIPA lysis method (see Section 4). 20 μg renal cell extract protein samples were resolved with SDS-PAGE and analyzed via Western blot for phosphorylated and total levels of (**A**) STAT1 and (**B**) STAT3 after CS or CS + RW. Data are the mean ± SEM (bars, *n* = 3) with representative blots displayed. Differences between group means were compared with 1-way ANOVA followed by Brown–Forsythe and Welch’s correction for multiple group comparisons; *p* < 0.05 was considered significant. *, *p* < 0.05; **, *p* < 0.01. Each solid circle, square, or triangle represent each experiment analysis (biological replicates) within the group.

**Figure 8 ijms-24-05468-f008:**
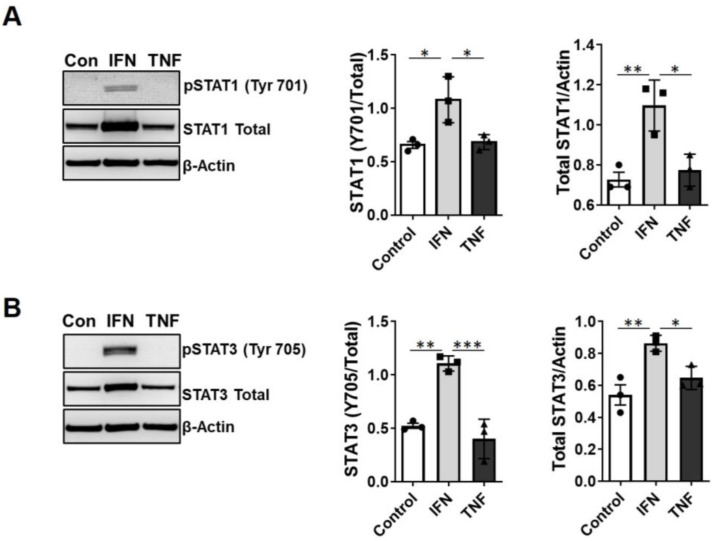
In vitro treatment of macrophages with IFN-γ increases phosphorylated or total levels of STAT1 or STAT3. (**A**,**B**) Raw 264.7 macrophages were exposed to IFN-γ (10 ng/mL) or TNF-α (10 ng/mL), and cell extracts were prepared using RIPA buffer (see Section 4). Proteins in macrophage extracts (20 μg) were resolved with SDS-PAGE and analyzed by Western blot for phosphorylated and total levels of (**A**) STAT1 and (**B**) STAT3 after IFN-γ or TNF-α treatment. Untreated cell extracts were used as controls. Data shown are mean ± SEM (*n* = 3) and these data underwent 1-way ANOVA analysis with Brown–Forsythe and Welch’s corrections for multiple comparisons; *p* < 0.05 was considered to be significant. *, *p* < 0.05; **, *p* < 0.01, ***, *p* < 0.001. Each solid circle, square, or triangle represent each experiment analysis (biological replicates) within the group.

**Figure 9 ijms-24-05468-f009:**
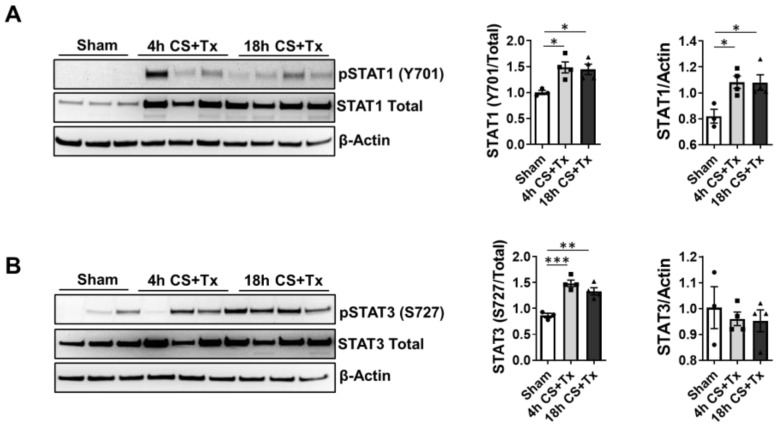
Phosphorylated and total STAT proteins remain elevated in renal allografts after 9 days post-surgery. Experimental groups reflect levels of CS exposure: Sham (no CS, right nephrectomy), 4 h CS + Tx (Fischer to Lewis rats), and 18 h CS + Tx (Fischer to Lewis rats). Surgical procedures were performed as described in Methods. (**A**,**B**) 30 μg of RIPA extracts prepared from kidney homogenates were subjected to SDS-PAGE and Western blot analysis of (**A**) STAT1 (Y701-phosphorylated and total protein) and (**B**) STAT3 (S727-phosphorylated and total protein) after Sham, 4 h CS + Tx, or 18 h CS + Tx. Data are the mean ± SEM (bars, *n* = 3), and representative blots are pictured. Differences between group means were compared with 1-way ANOVA followed by Brown–Forsythe and Welch’s corrections for multiple group comparisons; *p* < 0.05 was considered significant. *, *p* < 0.05; **, *p* < 0.01, ***, *p* < 0.001. Each solid circle, square, or triangle represent individual animals within the group.

## Data Availability

Not applicable.

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
