# Peer review of "Cold Storage Followed by Transplantation Induces Interferon-Gamma and STAT-1 in Kidney Grafts"

_ijms, 2023, doi:10.3390/ijms24065468_

Round 1

Reviewer 1 Report

The study aims to examine the impact of cold storage during kidney transplantation on the immunoregulatory roles of the signal transducer and activator of transcription (STAT) family proteins, STAT1 and STAT3. The data appear to be a continuation of the previous publication of the authors (PMID: 33998295). I have several concerns regarding the manuscript, as listed below:

Writing

1.     The authors should correct grammatical errors in the manuscript for clarity and readability.

2.     The use of "tyr" and "Y" in the Western blot in Figure 8 should be made more consistent for clarity.

3.     The representative figure for actin expression in Figure 9 is missing, which should be included.

4.     The method should specify that not only NRK cells, but also HK-2 cells, received treatment.

5.     The title should be revised to reflect the importance of STAT3, iNOS, and IL-12α, in addition to IFN-γ and STAT1, in kidney transplantation.

6.     The authors should clarify which results are relevant to the term “anti-inflammatory signaling” in the abstract description “STAT3 dephosphorylation was additionally observed after CS + transplant, a unique result suggestive of dysregulation of anti-inflammatory signaling and oxidative phosphorylation.”

to support the conclusion.

Method

1.     The authors mentioned that they performed 6 animal surgeries, The authors should explain why only 3 animals were used in the assessments, as this may affect the validity of the results.

Author Response

Major points:

  1. The authors should correct grammatical errors in the manuscript for clarity and readability.

Response: Thank you for the suggestion. The manuscript has been reviewed for grammatical errors by a native English speaker. You can find these corrections reflected in ‘Track Changes.’

  1. The use of "tyr" and "Y" in the Western blot in Figure 8 should be made more consistent for clarity. 

Response: We apologize for the error. Figure 8 has been updated for clarity as suggested. Additionally, a previously missed typo in Figure 9B was corrected.

  1. The representative figure for actin expression in Figure 9 is missing, which should be included.

Response: We now have included actin blot in the figure.

  1. The method should specify that not only NRK cells, but also HK-2 cells, received treatment.

Response: We apologize for not clarifying this before. Our methods have been updated as suggested. Our revised methods section 4.3.1 reads as follows:

NRK and HK-2 cells were exposed to CS at 70% confluence as reported previously [9]. CS condition cells were washed briefly in cold PBS (4°C) before being incubated with cold UW solution (4°C) for 18-h. Rewarming was initiated by washing the cells with cold PBS and then adding cold growth medium (4°C) before incubating cells in growth medium at 37°C for 6-h.

  1. The title should be revised to reflect the importance of STAT3, iNOS, and IL-12α, in addition to IFN-γ and STAT1, in kidney transplantation.

Response: Thank you for this. After carefully considering the responses of both reviewers to our title, we like to retain the current wording of our title because we have explored more effects of IFN -γ on STAT signaling. Also, Reviewer #2 appreciated the title: “I believe that the title is presented in a simple, clear, and concise way.”

  1. The authors should clarify which results are relevant to the term “anti-inflammatory signaling” in the abstract description “STAT3 dephosphorylation was additionally observed after CS + transplant, a unique result suggestive of dysregulation of anti-inflammatory signaling and oxidative phosphorylation.” to support the conclusion. 

Response: Thank you for bringing this to our attention. We have updated the wording in our abstract to bring clarity.

  1. The authors mentioned that they performed 6 animal surgeries, The authors should explain why only 3 animals were used in the assessments, as this may affect the validity of the results.

Response: We apologize for this error. We have corrected the typo “n = 6” to “n = 3” in all relevant methods sections. Our lab group is committed to minimizing our use of animals per study in accordance with guidelines by the NIH, and we have based our n = 3 value on previous power analyses using our CS+Tx model on proteasome function and kidney injury [PMID:  30303714].

Reviewer 2 Report

I would like to begin by greeting the authors and congratulating them for having decided to investigate an area where there is still so much to discover, but also for having decided to share this article with the rest of the scientific community, so that science can evolve.
This is a study on how cold storage followed by transplantation induces interferon-gamma and STAT-1 in kidney grafts. It is a very important topic, current and about which little is known.
All comments, doubts, and suggestions made are constructive and try to improve the article after several attentive readings.

Title
I believe that the title is presented in a simple, clear, and concise way.

Abstract
To what extent can this study change the practices of nephrologists around the world?

Key words:
Repetition of expressions that are in the title should be avoided. Whenever possible, the key words should be Mesh. Authors should review the keywords.

Introduction
The data presented in references 1 and 2 are from 2019. There is more recent data. And these data refer exclusively to the US population and not to the world population. If the authors decide to keep these numbers, this specific reference must be made.
Lines 36 to 42 do not have any bibliographic references.
Authors must indicate the objective at the end of the introduction, and this must be exactly the same as the abstract.
The introduction contains 40 bibliographic references. On such an up-to-date topic, I believe that the authors should make an effort to improve this component of the article, especially in the up-to-dateness of the references. Of these 40 references, only 9 are from 2017 or later, with 5 from 2019 and none from 2022. It feels like this part of the article was written and submitted like this in 2023, without any revision. The introduction deserves a little more effort on the part of the authors so that their work is enriched.
Also in the introduction, the authors must make clear reference to the reason why this article should be published in such an important journal as the International Journal of Molecular Sciences. What data from this article can change the practice of nephrologists around the world? What does this article add to scientific knowledge? Are there clinical practices that should be changed, or what kind of studies are still needed to prove these changes in humans?

Results
Results are presented in full, along with supplementary material that completes the information.

Discussion
Overall, I leave the same note that I left in the introduction regarding references.

Materials and methods
The type of study is not clearly indicated.
When was the data collected?
Line 406: "Transplant surgeries were performed as previously described", which implies that it was described in this article. It must be reviewed.
otherwise well described.

The Institutional Review Board Statement is not presented at the end of the article.

Author Response

Reviewer#2: "I would like to begin by greeting the authors and congratulating them for having decided to investigate an area where there is still so much to discover, but also for having decided to share this article with the rest of the scientific community, so that science can evolve.
This is a study on how cold storage followed by transplantation induces interferon-gamma and STAT-1 in kidney grafts. It is a very important topic, current and about which little is known.
All comments, doubts, and suggestions made are constructive and try to improve the article after several attentive readings."

Response: We appreciate your time and feedback.

Major Points:

  1. Abstract - To what extent can this study change the practices of nephrologists around the world?

Response: Thank you for the kind suggestion. Our abstract has been updated to include the following to better explain the significance of this study for nephrologists: Collectively, these results demonstrate that aberrant induction of STAT1 is sustained in vivo post-CS exposure and post-transplant. Thus, Jak/STAT signaling may be a viable therapeutic target during CS to mitigate poor graft outcomes when transplanting kidneys from deceased donors.”

  1. Key Words - Repetition of expressions that are in the title should be avoided. Whenever possible, the key words should be Mesh. Authors should review the keywords.

Response: We appreciate this suggestion. We have researched MeSH terms populated by the National Library of Medicine and we have updated the key words to be MeSH terms, as suggested.

  1. Introduction - On such an up-to-date topic, I believe that the authors should make an effort to improve this component of the article, especially in the up-to-dateness of the references. What data from this article can change the practice of nephrologists around the world? What does this article add to scientific knowledge? Are there clinical practices that should be changed, or what kind of studies are still needed to prove these changes in humans?

Response: We thank the reviewer for this suggestion, and we agree. The references in our introduction have been updated to reflect more recent data, and references have been added to Lines 36-42 as suggested. Statements clarifying the objective of the study have been added at the close of the introduction, and the potential use of targeting the Jak/STAT pathway therapeutically is emphasized.

  1. Materials and Methods - The type of study is not clearly indicated. When was the data collected?

Response: This is an animal study performed since 2018 and the data presented in this manuscript were obtained using the samples collected from 2018-2022.

  1. Line 406: "Transplant surgeries were performed as previously described", which implies that it was described in this article. It must be reviewed.

Response: Thank you for this. We have clarified our statement to direct the reader to our references, wherein our CS+Tx surgery model has been previously described.

  1. The Institutional Review Board Statementis not presented at the end of the article. 

Response: As our research does not involve human subjects, we have not added an Institutional Review Board Statement. Instead, our Institutional Animal Care and Use Committee (IACUC) statement is stated in the ‘Animals’ section of our methods to clarify that all animal experiments were performed in accordance with IACUC & NIH guidelines.

Round 2

Reviewer 2 Report

The authors of this study did a good job of responding to both reviewers. I am satisfied with the answers given and with the quality of the article presented.